# Toll-like Receptor Agonists Are Unlikely to Provide Benefits in Head and Neck Squamous Cell Carcinoma: A Systematic Review and Meta-Analysis

**DOI:** 10.3390/cancers15174386

**Published:** 2023-09-01

**Authors:** Sainiteesh Maddineni, Michelle Chen, Fred Baik, Vasu Divi, John B. Sunwoo, Andrey Finegersh

**Affiliations:** 1Division of Head and Neck Surgery, Department of Otolaryngology, School of Medicine, Stanford University, Stanford, CA 94305, USA; smaddineni@stanford.edu (S.M.); michelle.chen@stanford.edu (M.C.); fbaik@stanford.edu (F.B.); vdivi@stanford.edu (V.D.); sunwoo@stanford.edu (J.B.S.); 2Department of Otolaryngology, Veterans Affairs Palo Alto Health Care System, Palo Alto, CA 94304, USA

**Keywords:** immunotherapy, Toll-like receptor, head and neck cancer, meta-analysis, systematic review

## Abstract

**Simple Summary:**

Recurrent and metastatic head and neck cancer has limited treatment options and survival time is measured in months. Toll-like receptor agonists have been shown to improve tumor immune responses in preclinical studies and several clinical trials have now been performed. We performed a meta-analysis of existing clinical trials for recurrent and metastatic head and neck cancer and found there was no treatment benefit of these agents. While they do not appear to cause more adverse events, additional clinical trials may need to focus on new agents or drug combinations.

**Abstract:**

Background: Recurrent and metastatic (R/M) head and neck squamous cell carcinoma (HNSCC) has poor survival rates. Immunotherapy is the standard of care for R/M HNSCC, but objective responses occur in a minority of patients. Toll-like receptor (TLR) agonists promote antitumor immune responses and have been explored in clinical trials. Methods: A search for clinical trials using TLR agonists in HNSCC was performed under PRISMA guidelines. Data on patient characteristics, safety, and efficacy were collected and analyzed. Results: Three phase 1b trials with 40 patients and three phase 2 trials with 352 patients studying TLR8 and TLR9 agonists in combination with other treatment regimens for HNSCC were included. In phase 2 trials, there was no significant change in the objective response rate (RR = 1.13, CI 0.80–1.60) or association with increased grade 3+ adverse events (RR = 0.91, CI 0.76–1.11) associated with TLR agonist use. Conclusion: TLR agonists do not appear to provide additional clinical benefits or increase adverse events in the treatment of HNSCC. Given these results across multiple clinical trials and drug regimens, it is unlikely that additional trials of TLR agonists will demonstrate clinical benefits in HNSCC.

## 1. Introduction

Head and neck squamous cell carcinoma (HNSCC) arises from the upper aerodigestive tract and is the sixth most common malignancy worldwide [1]. HNSCC is treated with either surgical resection, radiation, chemotherapy, or a combination of these modalities, depending on its stage and location. This includes platinum-based chemotherapy in combination with cetuximab, especially in cases where a tumor is inoperable [2]. Based on studies using the SEER database, the 5-year survival for oral cavity and pharynx cancer was 68% and 61% for laryngeal cancer from 2012 to 2018 [3,4]. These survival rates have remained largely flat over the last several decades, despite improvements in surgical techniques, radiation delivery, and systemic therapies.

Over the last decade, immune checkpoint inhibitors (ICI) have emerged as an important treatment modality for HNSCC. These drugs inhibit the interaction between tumor and immune cells via the PD-1 and CTLA-4 receptors, thereby unmasking immunogenicity that leads to the increased immune cell infiltration of tumors. The Keynote-048 trial demonstrated that ICIs alone improve overall survival for recurrent and metastatic (R/M) HNSCC compared to cetuximab, establishing this regimen as the current standard of care for this treatment group [5]. Additionally, the combination of pembrolizumab with cetuximab has been potent [6]. Additional studies using ICIs in neoadjuvant and adjuvant settings are ongoing and likely to change treatment paradigms in the future. However, despite these successes, only about 20% of patients with HNSCC currently benefit from ICIs [7]. Considerable research is underway to identify pathways that may augment this response rate.

Toll-like receptors (TLRs) are key innate immune activating receptors capable of recognizing a set of conserved pathogens or damage-associated molecular patterns (PAMPs and DAMPs). TLRs can be expressed on the cell surface (TLR1, 2, 4, 5, 6) or on endosomal membranes (TLR3, 7, 8, and 9) [8]. In the context of cancer, antigen presenting cells, cancer cells, and other stromal cells can express TLRs and trigger inflammatory responses upon ligand binding [8]. TLR agonists are being increasingly evaluated as a therapeutic approach for cancer, as the engagement of TLRs can stimulate a pro-inflammatory cascade that could ultimately lead to greater tumor immune infiltration and the subsequent clearance of cancer cells or the potentiation of other immunotherapies. Here, we perform a meta-analysis of clinical trials of TLR agonists in HNSCC, including motolimod (TLR8 agonist), SD-101 (TLR9 agonist), IMO-2055 (TLR9 agonist), and EMD 1201081 (TLR9 agonist). For phase 1 trials we focus on evaluating the safety profile of the therapeutics, and for phase 2 trials we evaluate the efficacy of these agonists.

## 2. Materials and Methods

This systematic review was performed in accordance with the guidelines of the Preferred Reporting Items for Systematic Reviews and Meta-Analyses (PRISMA) [9]. The search term “Toll like receptors AND head and neck cancer” was used as a query to search Pubmed, Embase, and the Cochrane Library. References were also queried within selected articles. Inclusion criteria were clinical trials studying injections of TLR agonists, adult (>18 years of age) patients with HNSCC, at least 3 patients present in the study, adverse event reporting, clinical outcome reporting, and English language articles. The search was independently performed by two authors (SM and AF). This study was registered with the Open Science Framework (OSF) in accordance with PRISMA guidelines (https://osf.io/t8539).

Statistical analysis was performed in the R statistical software (version 4.2.1) environment. The Metafor statistical package was used for meta-analysis [10]. Relative risk ratios for objective responses and adverse events were generated, comparing studies with either placebo or low-dose TLR agonists to high-dose TLR agonists. A random effects model was used to generate an estimated average risk ratio for all studies. Relevant clinicodemographic variables were also extracted from the articles for presentation.

## 3. Results

### 3.1. Study Selection

From the initial 334 hits, we identified 20 candidate studies. Five studies were excluded because they were duplicates listed in multiple databases surveyed. Three studies were excluded because they were conference abstracts that were not yet peer reviewed. Three studies were excluded because they did not include HNSCC patients. Finally, three studies were excluded because they had HNSCC patients but failed to provide enough information about these patients’ baseline characteristics, the safety and tolerability of treatment, and/or the efficacy of the treatment. This left us with six final studies that were included in our analysis. Three studies were phase 1b trials of either motolimod or IMO-2055. The other three studies were phase 2 trials of motolimod, SD-101, or EMD 1201081. For these six studies, we collected data on patient characteristics, the safety of TLR agonists and adverse events related to treatment, and the efficacy of therapy. Figure 1 outlines the selection process of studies included in this systematic review [11].

### 3.2. Phase 1b Trials

We evaluated the safety of TLR agonists in HNSCC as reported in three phase 1b clinical trials. Two trials evaluated motolimod and one trial evaluated IMO-2055 [12,13,14]. All three trials evaluated these TLR agonists in combination with other agents, including cetuximab. Table 1 reports the baseline patient characteristics of the cohorts enrolled in these trials. The three trials enrolled 13–14 patients each. There were notable differences between these studies. Importantly, Shayan et al. studied patients with untreated HNSCC in a neoadjuvant systemic therapy trial prior to surgery, while the other two papers studied R/M HNSCC. There were also differences in tumor subsite distribution between the two motolimod studies, with Chow et al. having a greater oropharyngeal cancer representation and Shayan et al. having a larger proportion of oral cavity tumors. 

Additionally, studies in R/M HNSCC varied in their reporting of prior lines of treatment before trial enrollment. Chow et al. reported that 54% of patients had received one prior chemotherapy treatment, while 23% had received two or more prior chemotherapies. Machiels et al. noted that 100% of patients enrolled received prior curative treatment, but did not define how many lines of treatment patients received. Furthermore, Chow et al.’s paper showed that 92% of patients had distant metastases, while only 69% of patients in Machiels et al.’s study had distant metastases.

Table 2 summarizes key information about the safety of the TLR agonists based on adverse events (AEs) reported in each study. Motolimod was associated with one grade 3 or higher AE in Chow et al.’s study. Shayan et al.’s study did not report any grade 4 or 5 AEs associated with motolimod. In Machiels et al., 92% of patients receiving IMO-2055 experienced a grade 3+ AE, with one fatal AE and 31% of patients being discontinued from the study due to AEs. Given the significant toxicities associated with IMO-2055, the trial was terminated early. Overall, IMO-2055 was associated with a much poorer safety profile compared to motolimod amongst phase 1b trials.

Chow et al.’s study of motolimod and Machiels et al.’s study of IMO-2055 reported some efficacy data. For IMO-2055, 23% of patients reported a partial response and 0% had a complete response. For Chow et al.’s motolimod study, 15% of patients had a partial response and 0% had a complete response. Due to it being a neodjuvant trial, Shayan et al.’s motolimod study did not report efficacy data (Figure 2).

### 3.3. Phase 2 Trials

Three phase 2 trials evaluating SD-101 (Cohen et al.), motolimod (Ferris et al.), and EMD 1201081 (Ruzsa et al.) in patients with R/M HNSCC were included in our analysis [15,16,17]. Table 3 outlines the baseline patient characteristics of these studies in patients. Cohen et al. included comparisons between low dose (2 mg/lesion) and high dose (8 mg/lesion) groups, while Ferris et al. and Ruzsa et al. reported comparisons to a placebo group that did not receive a TLR agonist. In Cohen et al.’s study, 24% of patients did not receive any prior systemic therapy, while 35% of patients in Ferris et al.’s study did not. In Ruzsa et al.’s study, all but one patient had prior chemotherapy. Only 37% of patients in Ruzsa et al.’s study had metastases, while 92% had them prior to enrollment for Cohen et al.’s study. Ferris et al.’s study did not report how many patients had metastatic cancer.

In Cohen et al.’s SD-101 study, more grade 3+ AEs occurred in the cohort receiving the 8 mg dose (34.8%) compared to the 2 mg dose (14.8%). In Ferris et al.’s motolimod study, similar rates of AEs were reported and fatal AEs occurred in both the treatment and placebo groups. Finally, in Ruzsa et al.’s EMD 1201081 study, patients were discontinued due to AEs in both treatment and placebo cohorts (Table 4). We pooled these studies to evaluate the AE rates in the placebo or low-dose group (*n* = 176) compared to the treatment group (*n* = 176). There was no significant difference in the relative risk of grade 3+ AEs between these groups (RR 0.91, 95% CI 0.76–1.11) (Figure 3).

Efficacy data were reported by all phase 2 trials (Table 5). Objective response rates (ORR) in the treatment groups in Ferris et al.’s motolimod study and Ruzsa et al.’s EMD 1201081 study were not significantly different from the placebo group. The median PFS in the 2 mg cohort of SD-101 in Cohen et al.’s study was 2.5 months, vs. 2.3 months in the 8 mg cohort. PFS in the treatment vs. the placebo group for Ferris et al.’s motolimod study was 6.1 vs. 5.9 months. Finally, PFS in the treatment vs. the placebo group for Ruzsa et al.’s EMD 1201081 study was 1.5 vs. 1.9 months. We pooled the ORR in these studies to compare the relative risk of response in the placebo or low-dose group (*n* = 176) to the treatment group (*n* = 176) There was no significant difference in ORR between these groups (RR 0.93 95% CI 0.60–1.44) (Figure 4). Lastly, funnel plots of grade 3+ AEs and ORR between groups in these studies did not show any evidence of publication bias in these phase 2 trials (data not shown).

## 4. Discussion

In this meta-analysis, we evaluate the safety and efficacy of TLR agonists reported in various clinical trials of patients with HNSCC. We pooled 40 patients in phase 1b and 176 patients in phase 2 trials receiving TLR agonists in combination with other systemic therapies to identify both adverse event rates and treatment efficacy. Based on currently available studies, it does not appear that the addition of TLR agonists to standard-of-care regimens for R/M HNSCC provides treatment benefits or higher rates of grade 3+ AEs.

While TLR agonists have demonstrated efficacy in preclinical studies [18,19,20,21], this meta-analysis demonstrates that they may not have a role for the management of R/M HNSCC. There are several possible reasons for the lack of efficacy across three phase 2 trials. Importantly, the majority of patients with R/M HNSCC had previously undergone radiation therapy, which can act as a double-edged sword to promote systemic tumor antigen presentation to immune effectors, but can also be partly immunosuppressive in a post-treatment setting [22,23]. The extent to which this interplay occurs likely varies between patients. In one of the phase 2 trials included in this study, Cohen et al. performed transcriptomic analysis of tumors pre- and post-injection and showed significantly enhanced CD8+ T cell and NK cell gene expressions within tumors of responders, but no difference in non-responders. Additionally, there was no difference in response rates for patients with high and low PD-L1 expression in this study. However, whether this response relates to the TLR agonist or pembrolizumab given concurrently is not known, since neither were tested separately. But, based on the ineffectiveness of TLR agonists in other studies, is likely to relate to ICIs that are now the standard of care for R/M HNSCC patients. 

Shayan et al. was unique among the studies included in this meta-analysis in that TLR agonist was given to treatment-naïve patients in a neoadjuvant setting in combination with cetuximab. This study also evaluated pre- and post-treatment tumor samples for tumor immune infiltration and found that motolimod plus cetuximab was associated with a decreased induction of Tregs and enhanced CD8+ T cell infiltration of tumors following treatment. While this treatment-naïve group offers a better model for TLR-driven immune infiltration, this study was again limited by the lack of a control group as Cetuximab is associated with similar effects on the tumor-immune microenvironment [24]. Despite limitations of these studies, the mechanisms of TLR-agonist stimulation in clinical trials are unlikely to have relevance if they lack a treatment benefit. 

The rate of grade 3+ AEs ranged from 8% to 92% in each cohort, with Machiels et al.’s study of IMO-2055 notably being terminated early due to safety concerns. Fatal AEs were uncommon but were reported in Machiels et al.’s IMO-2055 study and Ferris et al.’s motolimod study. Since these patients were receiving cetuximab, ICI, and other systemic therapies that have high rates of grade 3+ AEs, the contribution of TLR agonists to overall AE reporting is unclear. Importantly, pooling AE rates in phase 2 trials did not show an increased risk of AEs in the TLR agonist group. Looking at AEs reported for these TLR agonists in other trials, in a trial of SD-101 in melanoma, 27% of patients had a grade 3–4 AE related to SD-101, 41% had a serious adverse event, and no patient had dose-limiting toxicities or death [25]. A phase 2 trial of motolimod in ovarian cancer also did not identify severe toxicities leading to treatment discontinuation, but did identify a serious adverse event in 40.8% of patients in both the placebo and motolimod groups [26]. In a trial of IMO-2055 in non-small cell lung cancer, 34% had a grade 3+ AE and 11% had serious AEs related to IMO-2055 [27]. These studies, in combination with the data presented in this trial, suggest that TLR agonists do not add a significant risk for severe AEs beyond toxicities associated with standard-of-care therapies.

There have been numerous clinical trials of TLR agonists in cancers beyond HNSCC [28]. SD-101 is being tested in combination with anti-PD-1 and/or additional agents in pancreatic, prostate, breast, uveal melanoma, and hepatic or other solid tumors [28]. TLR7/8 agonist NKTR-262 increased CD11c dendritic cell recruitment to tumors in melanoma, but a trial of NKTR-262 in combination with bempegaldesleukin with or without nivolumab was terminated. BDB001 and CV8102 are other TLR7/8 agonists being investigated in solid tumors and melanoma, respectively [29]. Additionally, lefitolimod and cavrotolimod are TLR9 agonists being evaluated in solid tumors, while tilsotolimod and vidutolimod/CMP-001 are TLR9 agonists being tested in melanomas [29]. As additional trial data are published for TLR agonists in HNSCC, evidence that demonstrates clinical benefits in conjunction with other treatments may surface that may warrant a re-examination of previously reported clinical trials.

It is important to note limitations of this study in evaluating future clinical trials for TLR agonists. Notably, only 176 patients from phase 2 trials were included and treatment modalities varied between trials. This is consistent with current clinical practice, where systemic therapy regimens are not consistent among patients with R/M HNSCC due to differences in performance status and comorbidities that may limit a patient’s ability to receive platinum-based chemotherapy or cetuximab. Now that the Keynote-048 study has shown that pembrolizumab alone is efficacious in this patient group, a clinical trial design similar to Cohen et al. but comparing ICIs alone to ICIs with TLR agonists seems best powered to definitively ascertain their clinical benefit. If TLR agonists are not found to have a benefit in well-powered clinical trials, studying other in situ vaccines, like oncolytic viruses, may also determine whether injection-based strategies aimed at unlocking intratumoral immune responses are worth pursuing.

## 5. Conclusions

In this systematic review, we evaluate the safety and efficacy of TLR agonists reported in various phase 1b and phase 2 trials of HNSCC. Overall, TLR agonists are generally tolerable and not associated with fatal or therapy-terminating toxicities. However, TLR agonists in phase 2 trials typically did not show significant clinical benefits. TLR agonists thus do not appear to be highly appealing as a future therapeutic avenue for immunotherapy in HNSCC.

## Figures and Tables

**Figure 1 cancers-15-04386-f001:**
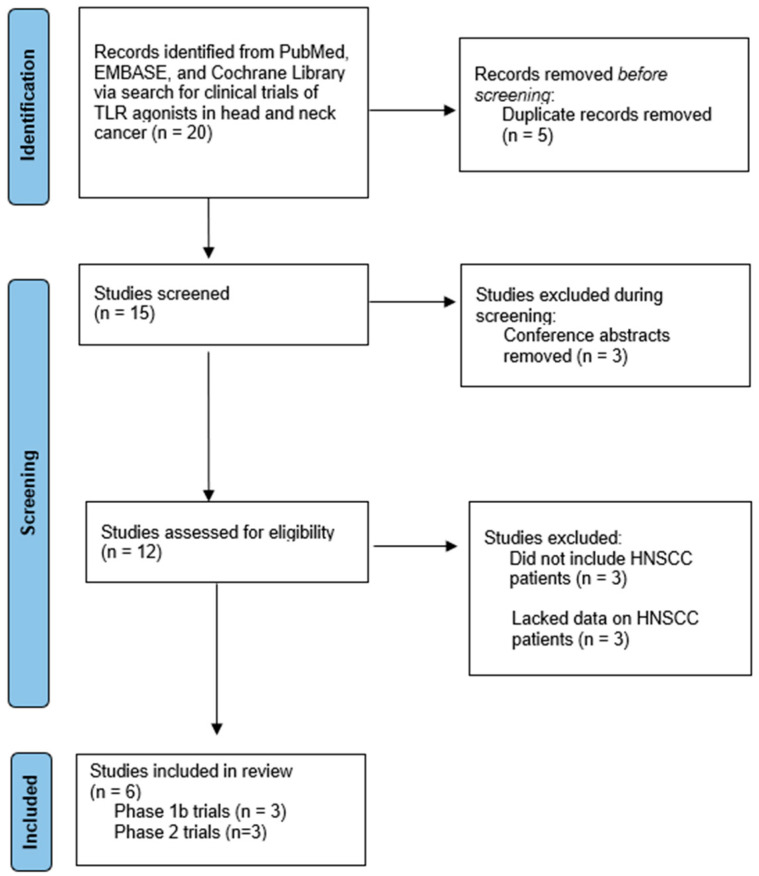
PRISMA flow diagram of study screening and inclusion.

**Figure 2 cancers-15-04386-f002:**
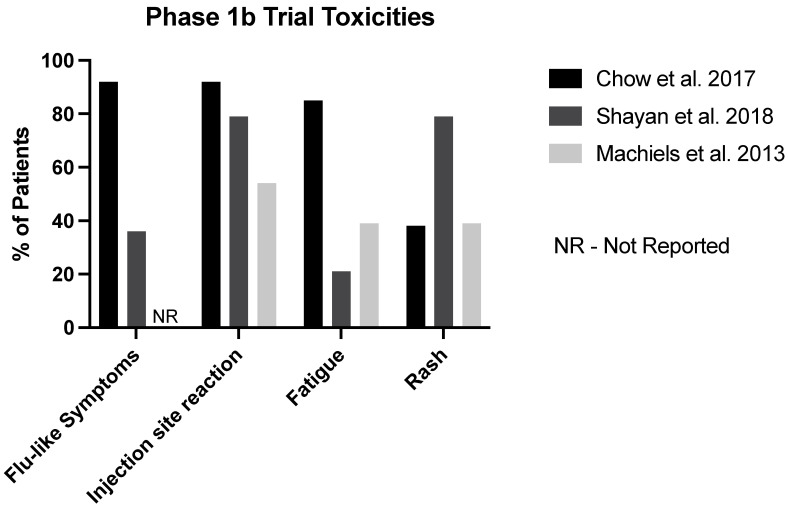
Summary of common toxicities noted in phase 1b trials [12,13,14].

**Figure 3 cancers-15-04386-f003:**
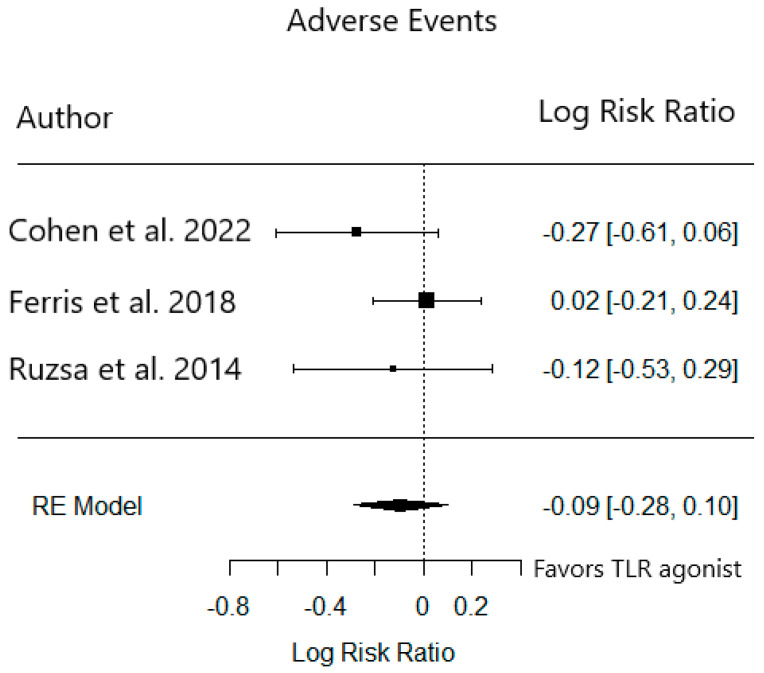
Relative risk of grade 3+ adverse events in phase 2 trials [15,16,17].

**Figure 4 cancers-15-04386-f004:**
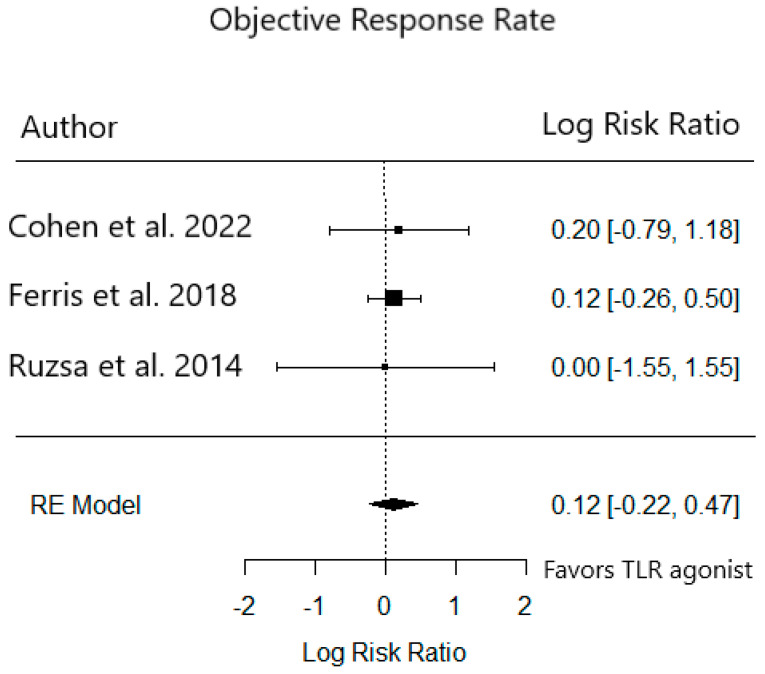
Relative risk of objective response to TLR agonist in phase 2 trials [15,16,17].

**Table 1 cancers-15-04386-t001:** Baseline cohort characteristics for phase 1b trials.

Study	Chow et al. 2017 [12]	Shayan et al. 2018 [13]	Machiels et al. 2013 [14] *
**Agent**	Motolimod	Motolimod	IMO-2055
**Treatment**	Motolimod + cetuximab	Motolimod + cetuximab	IMO-2055 + 5-fluorouracil, cisplatin, and cetuximab
**Patient Population**	R/M HNSCC	Untreated HNSCC	R/M HNSCC
**Number of Patients**	13	14	13
**Median Age**	62	61	59
**ECOG**			
0 (%)	15	-	62
1 (%)	70	-	38
≥2 (%)	15	-	0
**Sex**			
Male (%)	77	64	92
Female (%)	23	36	8
**Tumor Site**			
Oral cavity (%)	15	71	38
Oropharynx (%)	46	7	38
Larynx (%)	23	14	8
Hypopharynx (%)	8	7	15
Other (%)	8	0	0
**HPV Status**			
Positive (%)	23	-	-
Negative (%)	8	-	-
Unknown (%)	70	-	-
**Prior Treatment**			
Chemotherapy (%)	77	-	69
Radiation (%)	92	-	100
Surgery (%)	70	-	77
Cetuximab (%)	77	-	-
**Recurrence Type**			
Locoregional (%)	8	-	31
Distant Metastasis (%)	46	-	69
Both (%)	46	-	0

* Trial terminated early for safety concerns.

**Table 2 cancers-15-04386-t002:** Adverse events in ≥20% of patients in phase 1b trials.

Study	Chow et al. 2017 [12]	Shayan et al. 2018 [13]	Machiels et al. 2013 [14] *
**Agent**	Motolimod	Motolimod	IMO-2055
**Treatment**	Motolimod + cetuximab	Motolimod + cetuximab	IMO-2055 + 5-fluorouracil, cisplatin, and cetuximab
**Number of Patients**	13	14	13
**Dosage**	Motolimod: 2.5 mg/m^2^, 3.0 mg/m^2^, or 3.5 mg/m^2^ Cetuximab: 250 mg/m^2^	Motolimod: 2.5 mg/m^2^ Cetuximab: 400 mg/m^2^ loading then 250 mg/m^2^	IMO-2055: 0.16 mg/kg or 0.32 mg/kg Cetuximab: 400 mg/m^2^ loading then 250 mg/m^2^ Cisplatin: 100 mg/m^2^/day 5-fluorouracil: 1000 mg/m^2^/day
**Flu-Like Symptoms (%)**	92	36	-
**Injection Site Reaction (%)**	92	79	54
**Fatigue (%)**	85	21	39
**Rash (%)**	38	79	39
**Grade 3+ AEs (%)**	8	- *	92
**Fatal AEs (%)**	0	0	8
**Discontinued Due to AEs (%)**	0	0	31

* No grade 4 or 5 AEs reported.

**Table 3 cancers-15-04386-t003:** Baseline cohort characteristics for phase 2 trials of R/M HNSCC.

Trial Reference	Cohen et al. 2022 [15]		Ferris et al. 2018 [16]		Ruzsa et al. 2014 [17]	
**Agent**	SD-101		Motolimod		EMD 1201081	
**Treatment Group**	SD-101 8 mg + pembrolizumab	SD-101 2 mg + pembrolizumab	EXTREME regimen + motolimod	EXTREME regimen + placebo	EMD 1201081 + cetuximab	Cetuximab only
**Number of Patients**	23	28	100	95	53	53
**Median Age**	65	63	58	60	58	57
**ECOG**						
0 (%)	26	18	38	39	23	23
1 (%)	74	82	62	61	77	77
≥2 (%)	0	0	0	0	0	0
**Sex**						
Male (%)	91	68	85	85	85	85
Female (%)	9	32	15	15	15	15
**Tumor Site**						
Oral cavity (%)	57	46	27	27	-	-
Oropharynx (%)	9	32	40	45	-	-
Larynx (%)	17	11	22	21	-	-
Hypopharynx (%)	0	7	4	5	-	-
Other (%)	17	4	7	1	-	-
**HPV Status**						
Positive (%)	26	36	60 *	65 *	-	-
Negative (%)	26	39	33 *	28 *	-	-
Unknown (%)	48	25	8 *	7 *	-	-
**Prior Treatment**						
Chemotherapy (%)	-	-	63	58	98	100
Radiation (%)	87	75	79	85	85	77
Surgery (%)	96	86	56	56	38	53
Cetuximab (%)	-	-	10	21		
**Recurrence Type**						
Locoregional (%)	9	7	-	-	70	57
Distant Metastasis (%)	61	57	-	-	30	43
Both (%)	30	36	-	-	-	-

* HPV status reported for oropharyngeal tumors only.

**Table 4 cancers-15-04386-t004:** Adverse events in patients in phase 2 trials.

Trial Reference	Cohen et al. 2022 [15]		Ferris et al. 2018 [16]		Ruzsa et al. 2014 [17]	
**Agent**	SD-101		Motolimod		EMD 1201081	
**Treatment Group**	SD-101 8 mg + pembrolizumab	SD-101 2 mg + pembrolizumab	EXTREME regimen + motolimod	EXTREME regimen + placebo	EMD 1201081 + cetuximab	Cetuximab only
**Number of Patients**	23	27	86	86	54	53
**Dosage**	SD-101 8 mg in 1 lesion + pembrolizumab	SD-101 2 mg in 1–4 lesions + pembrolizumab	Motolimod 3 mg/m^2^ + cisplatin 100 mg/m^2^ + fluorouracil 1000 mg/m^2^ + cetuximab 400 mg/m^2^ loading then 250 mg/m^2^	Placebo + cisplatin 100 mg/m^2^ + fluorouracil 1000 mg/m^2^ + cetuximab 400 mg/m^2^ loading then 250 mg/m^2^	EMD 1201081 0.32 mg/kg + cetuximab 400 mg/m^2^ loading then 250 mg/m^2^	Cetuximab 400 mg/m^2^ loading then 250 mg/m^2^
**Chills**	44	11	37	6	-	-
**Pyrexia**	26	22	43	12	19	6
**Injection Site Reaction**	17 *	4 *	39	0	20	0
**Fatigue**	74	56	43	45	15	23
**Rash**	-	-	19	27	30	32
**Grade 3+ AEs (%)**	35	15	39	40	56	51
**Fatal AEs (%)**	0	0	5	8	0	0
**Discontinued Due to AEs (%)**	-	-	-	-	19	15

* Injection site reaction not directly reported, so injection site erythema used as a proxy.

**Table 5 cancers-15-04386-t005:** Efficacy data for phase 2 trials of R/M HNSCC.

Trial Reference	Cohen et al. 2022 [15]		Ferris et al. 2018 [16]		Ruzsa et al. 2014 [17]	
**Agent**	SD-101		Motolimod		EMD 1201081	
**Treatment Group**	SD-101 8 mg + pembrolizumab	SD-101 2 mg + pembrolizumab	EXTREME regimen + motolimod	EXTREME regimen + placebo	EMD 1201081 + cetuximab	Cetuximab only
**Number of Patients**	28	23	100	95	53	53
**Median Age**	63	65	58	60	58	57
**ORR (%)**	21	26	38	34	6	6
CR (%)	7	0	2	5	0	0
PR (%)	14	26	36	28	6	6
SD (%)	25	22	22	24	32	38
PD (%)	36	39	9	8	40	34
**Median PFS**	2.5	2.3	6.1	5.9	1.5	1.9
**Median OS**	Not reached	9	13.5	11.3	6.3	-

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
