# Peer review of "Toll-like Receptor Agonists Are Unlikely to Provide Benefits in Head and Neck Squamous Cell Carcinoma: A Systematic Review and Meta-Analysis"

_cancers, 2023, doi:10.3390/cancers15174386_

Round 1
Reviewer 1 Report
Maddineni et al. performed a meta-analysis of the toll-like receptor agonists in clinical trials of head and neck squamous cell carcinoma patients. They reviewed 6 studies in phases 1b and 2 trials. They concluded that TLR agonists do not appear to provide additional clinical benefit in HNSCC patients. Although the limited number of studies included in this study, the review describes the effect of TLR agonists on the currently available studies.
Author Response
We thank the reviewer for their positive comments and feedback. We have made major changes to the manuscript based on suggestions from all reviewers and feel they have improved the text.
Reviewer 2 Report
To Author:
Most head and neck cancers are derived from the mucosal epithelium in the oral cavity, pharynx and larynx, and are known collectively as head and neck squamous cell carcinoma (HNSCC). In this paper, the authors analyzed the safety and efficacy of TLR (Toll-like receptor) agonists in the treatment of recurrent and metastatic head and neck squamous cell carcinoma through a meta-analysis. I considered this paper lacks sufficient innovation and research significance. In addition, some of the analyses were not sufficient as detailed below.
Comments:
(1) In the introduction, the current clinical treatments of HNSCC were not introduced in detail.
(2) The number of patients with metastases was not marked in the table.
(3) Most of the HNSCC patients analyzed were male, but whether gender affects the efficacy of TLR agonists in treating HNSCC is not clear. The author did not analyze and discuss this effect.
(4) There are different types of HNSCC (such as HPV-negative and HPV-positive HNSCC), and the authors did not consider the influence of HNSCC classification on the results.
Author Response
We thank the reviewer for their positive comments and feedback. We have made major revisions to the manuscript based on suggestions from all reviewers and feel they have improved the text.
- In the introduction, the current clinical treatments of HNSCC were not introduced in detail.
- We have added additional references regarding treatment options for HNSCC to the text (lines 37-38 and 49-50)
- The number of patients with metastases was not marked in the table.
- We have specified patients with distant metastasis in Tables 1 and 3.
- Most of the HNSCC patients analyzed were male, but whether gender affects the efficacy of TLR agonists in treating HNSCC is not clear. The author did not analyze and discuss this effect.
- We specified the number of patients that were male and female in Table 1. Unfortunately, because this is a meta-analysis, we are limited by the data provided by the original articles, which did not analyze sex differences. Therefore, it is not possible to study this effect due to limitations with existing Phase 1 and 2 clinical trials.
- There are different types of HNSCC (such as HPV-negative and HPV-positive HNSCC), and the authors did not consider the influence of HNSCC classification on the results.
- Data relating to HPV status was only presented in studies that had a significant number of patients with oropharyngeal malignancies. Where this information was available, it was included in Tables 1 and 3.
- All patients in phase 2 trials had locoregionally advanced disease that was unresectable or distant metastases (or both) and therefore were Stage IV.
Reviewer 3 Report
In their study, Maddineni and colleagues investigated the potential efficacy of TLR-agonists for managing head and neck squamous cell carcinoma (HNSCC). They executed a systematic review and meta-analysis of existing phase 1b and phase 2 clinical trials in line with the PRISMA guidelines. The data harvested from these trials were harnessed to assess the safety and performance of TLR agonists. The evaluations from these clinical trials revealed that although TLR agonists were generally bearable and did not correlate with fatal or treatment-halting side effects, they exhibited no significant clinical advantage in phase 2 trials. Thus, the authors inferred that TLR agonists are unlikely to yield substantial benefits in HNSCC treatment.
Nonetheless, for the final acceptance of the article, the authors need to address several concerns:
- The review is anchored on a limited selection of clinical trials, which might not offer a holistic understanding of the role of TLR agonists in HNSCC treatment. A broader range of studies and clinical trials is necessary to validate these findings.
- The authors could enhance the presentation of data by incorporating charts and graphs for a more straightforward visualization of the key findings and data trends.
- The authors need to discuss the limitations of their review and the studies included in it.
- Given the potential lack of benefit of TLR agonists in HNSCC treatment, the authors should provide their insights and recommendations for other potential therapeutic targets or strategies for future exploration.
- Although the references are cited within the text, a comprehensive list of references is absent at the end of the article.
Author Response
We thank the reviewer for their feedback and generally positive comments regarding the manuscript. We have made major revisions to the text and addressed the suggestions you made. We feel they have strengthened the text.
- The review is anchored on a limited selection of clinical trials, which might not offer a holistic understanding of the role of TLR agonists in HNSCC treatment. A broader range of studies and clinical trials is necessary to validate these findings.
- We agree that the overall number of studies is limited but the aggregate number of patients in phase 2 trials (n = 176) is reasonable to power a difference in objective response rates, which was not seen. We have added a paragraph in the discussion acknowledging this limitation (lines 245-256).
- The authors could enhance the presentation of data by incorporating charts and graphs for a more straightforward visualization of the key findings and data trends.
- We have added a figure (Figure 2) to graphically summarize the adverse effects data presenting in the tables.
- The authors need to discuss the limitations of their review and the studies included in it.
- Thank you for pointing out this deficiency in the text. We have added an extended paragraph on key limitations to the paper (lines 245-256)
- Given the potential lack of benefit of TLR agonists in HNSCC treatment, the authors should provide their insights and recommendations for other potential therapeutic targets or strategies for future exploration.
- Thank you for pointing out this deficiency in the text. We have added suggestions for new therapeutic targets (lines 254-256)
- Although the references are cited within the text, a comprehensive list of references is absent at the end of the article.
- This is included at the end of our manuscript. There may be a formatting issue if this is not available to view while reviewing the manuscript – we can discuss with the journal staff if there is an issue.
Reviewer 4 Report
This study is interesting with clinical significance. Toll-like receptor agonists is a potential antitumor drug . There are some studies on the efficacy of toll-like receptor agonists to recurrent to metastatic head and neck cancer. The authors made an excellent review with a strong focus and comprehensive scientific evidences on that . The followings are some comments to the authors.
Comments
1.Cause previous treatment is an important factor affecting the efficacy, I suggest adding the number of front-line treatment such as first line, second line and the number of organ metastasis in Table 1.
2.I suggest adding a table about treatment related adverse event (TRAE) like the form of table 2.
3.Whether PD-L1 expression was detected in SD-101 study and EGFR expression in EMD 1201081? I recommend adding the information above in the baseline.
Author Response
We thank the reviewer for their feedback and generally positive comments regarding the manuscript. We have made major revisions to the text and addressed the suggestions you made. We feel they have strengthened the text.
1.Cause previous treatment is an important factor affecting the efficacy, I suggest adding the number of front-line treatment such as first line, second line and the number of organ metastasis in Table 1.
- We have included prior treatments and distant metastases in Table 1. The site of distant metastasis was not available within the clinical trial data of the included papers.
2.I suggest adding a table about treatment related adverse event (TRAE) like the form of table 2.
- We have made the treatment related adverse event data easier to interpret by including a new figure to represent the most common AEs. Please see Figure 2.
3.Whether PD-L1 expression was detected in SD-101 study and EGFR expression in EMD 1201081? I recommend adding the information above in the baseline.
- This is important information to consider and is valuable for stratifying patients that are receiving treatment. Unfortunately, EGFR expression is not reported in either the EXTREME OR EMD 1201081 trials. PD-L1 expression is reported in the in the SD-101 study and we have added text to the discussion to indicate that the authors found no significant difference in objective response rates based on PD-L1 expression in this study (lines 202-204).
Round 2
Reviewer 2 Report
The revised paper is OK. I don't have any comments.
Reviewer 3 Report
The authors have addressed all the comments satisfactorily.